# Portuguese Agrifood Sector Resilience: An Analysis Using Structural Breaks Applied to International Trade

**Maria de Fátima Oliveira** [1,2,*] and **Pedro Reis** [3]

1 Polytechnic of Coimbra, Coimbra Agriculture School, Bencanta, 3045-601 Coimbra, Portugal
2 Research Centre for Natural Resources Environment and Society (CERNAS), Escola Superior Agrária de Coimbra, Escola Superior Agrária de Coimbra, Bencanta, 3045-601 Coimbra, Portugal
3 National Institute of Agricultural and Veterinary Research, I.P, Quinta do Marques, 2780-157 Oeiras, Portugal; pedro.reis@iniav.pt
* Correspondence: foliveira@esac.pt

**Abstract:** In the last two decades, Portugal suffered the effects of two global crises, the financial crisis and the COVID-19 pandemic, as well as the Common Agriculture Policy reforms. These crises had a great impact on the Portuguese economy, but it is completely unclear how they affected the dynamics of the Portuguese agrifood sector. This study's objective is to analyze the resilience of this sector to European and global socks, testing the effects on international trade. Secondary data from the Portuguese Statistics Institute were used for the exports and imports trade series of animal and vegetable products and food industries from 2000 to 2020. The methodology was based on the structural xtbreak model, stability analysis, and tests for structural breaks. Some volatility was observed in the trade series, particularly in imports, without consistency among years, trade sectors, or imports versus exports trade. In the case of exports, one or two structural breaks in the different sectors occurred in different years. The most relevant dynamics occurred after the sovereign debt crisis. It was concluded that CAP reforms and global crises seem to not have caused new relevant dynamics in the Portuguese international agrifood trade. This revealed the resilience of the sector to external shocks.

**Keywords:** agrifood trade; Portuguese agriculture; structural breaks; xtbreak STATA

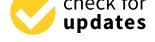



## 1. Introduction

The recent COVID-19 pandemic has revealed public concerns about food production and the capacity of supply chains during catastrophic events. This public awareness of the importance of agriculture and food supply chains was heightened by the war in Ukraine. Several actors have highlighted the strategic importance of the food supply chains, the competitive advantages of the companies that deliver products, and the collaborative relationships between the links, revising the definition of the value of chains' sustainability and presenting a methodology to analyze its performance [1–5]. Thus, it is important to understand their dynamics in the context of an epidemic outbreak or other situations of national or international shock. Governments need to better understand these types of shocks and the impacts on food chains to design strategies to guarantee conditions to keep the agrifood complex functioning, guaranteeing the security of food supply in a context of strong movement restrictions for people and goods, and mitigate the effect of a drop in demand for the various sub-sectors.

Several works have been carried out to observe the effects of the sovereign debt crisis (originated by the subprime crisis in 2008) and the COVID-19 crisis on international trade. OECD [6] highlights the heterogeneity of the pandemic in trade regarding products and trade destinations, which can mean uncertainty and create pressure on some sectors. In recent years, relevant studies have been carried out in various sectors, including both social and economic analyses, with an emphasis on studies analyzing the effect of COVID-19 on

the population, namely mortality and morbidity [7–9]. Despite the short time between the beginning of the COVID-19 pandemic and now, there is already an important number of studies on the effects of COVID-19 on the agricultural industry. OECD [6] and FAO [10] work highlighted the rapid recovery of the agrifood sector after the pandemic. Erol and Saighain [11] presented a brief bibliographical analysis of the effect of the pandemic, namely the effect on prices and distortions in the food chain. The empirical analysis of this work highlights the resilience of the US meat sector. Other works were carried out on structural breaks, but they did not focus on pandemic crises but on the effect of economic crises on economic series [12–14].

Portugal has a high agrifood trade deficit: 3581.3 € (average of the last three assessable years between 2018 and 2020) [15]. About 50% corresponds to agriculture, and the other 50% to the food, beverage, and tobacco (FB&T) industries. Agriculture accounts for 1.6% of exports and 3.7% of imports (average of the last three years), while FB&T industries account for 6.4% of exports and 8.4% of imports (ibid). The latest available data for the degree of self-sufficiency are from 2018. The average for the three-year period 2016–2018 is 85.4%. The rate of coverage of imports by exports (average between 2018 and 2020) was 75.3% for FB&T industries but only 43.1% for agriculture. This reality of the agrifood sector, and of agriculture in particular, has major differences between the various production sectors. For example (average of the last 10 years) (ibid), wine and olive oil had a positive balance of 550 M€ and 42 M€, respectively (coverage rates of 631% and 93%), while the trade balances of cereals and meat were −615 M€ and −688 M€, respectively (coverage rates of 7% and 15%).

Portugal was hit hard and was subject, between 2011 and mid-2014, to external financial assistance through the granting of an external loan of €78 billion from the European Community (EC), International Monetary Fund (IMF), and European Central Bank (ECB) in exchange for the application of a very severe austerity program. This set of highly restrictive budgetary measures led to a contraction of the economy during the 2011–2013 period, with Gross Domestic Product (GDP) in volume decreasing by 6.9%, in year-on-year accumulated terms, in this period, leading to an increase to historic levels of the unemployment rate (16.2%, on average, in 2013). Along with a decrease in employment in Portugal for five consecutive years (−13%, accumulated year-on-year terms, between 2009 and 2013) and a large flow of emigration, only comparable to that of the 1960s [16–18].

The COVID-19 pandemic and the necessary containment measures have caused a very severe shock to Portuguese businesses, especially in accommodation and food services. According to the results of a fast and exceptional enterprise survey applied to Portugal to characterize the short-term economic impact of the pandemic [19]. It was the sector with the longer average closure duration and greater short-term impacts on jobs. But agriculture could not stop and kept working, had positive behavior, and proved to be resilient and capable of innovating, adapting, and reinventing itself in the face of the existing constraints [20]. Portugal, like many other countries, had a big drop in GDP in 2020, −8.3% [21], and a strong impact on foreign trade (of goods and services): −18.6% in exports and −11.8% in imports (ibid). Policy measures were implemented to avoid company insolvencies, ensuring the maintenance of the productive structure, and the impact on households (unemployment and default on credit payments). In 2021 and 2022, there was a recovery in GDP and foreign trade. Exports grew by 13.4% and 16.6%, and imports by 13.2% and 11.1% (ibidem).

Due to the inelasticity of the demand in the agrifood sector, international food trade is an excellent indicator to analyze the sustainability and resilience of the agriculture sector and agroindustry. We will seek to identify disruptions in foreign trade patterns, which factors may be associated with these outbreaks (domestic or international shocks), and the level of impact on international food trade. The Portuguese case is interesting because it is very dependent on international food supply markets, was very affected by the sovereign debt crisis, and experienced a strong increase in some agricultural products' exportation. Our study has two aims: to understand the better methodology to identify the

international food trade outbreaks and the possible impacts of the sovereign debt crises and the COVID-19 pandemic on the Portuguese food trade balance. This work analyzes the structural breaks in a long series of temporal data from 2000 to 2020.

The article uses the xtbreak model and stability tests. A structural break usually refers to a significant shift or alteration in the underlying dynamics of a time series and can have important implications for understanding patterns, trends, and relationships in data. It also adds knowledge about the effects of economic and non-economic events on international trade in a small country that is exposed to international trade. Regarding stationarity, this issue is debated because stationarity is a statistical property of time series data where statistical properties such as mean and variance remain constant over time. Stationarity is an essential assumption in many time series analysis techniques. If the article focuses on stationarity issues in the context of agrifood trade in Portugal, it should indeed be explicitly stated in the Introduction or Abstract to provide readers with a clear understanding of the scope and objectives of the article.

## 2. Materials and Methods

The methodologies that have been applied to analyze structural breaks in different scientific works, regardless of the object under analysis, have been distinct. Some authors, such as Karavias et al. [22], applied methodologies to panel data. Other authors performed a statistical analysis through correlations to assess the effects of the pandemic on the markets [23]. Barbero et al. [24] conducted a bibliographical analysis of the effects of the COVID-19 pandemic on international trade and applied a gravity model to assess the effects of the pandemic on international trade using monthly series. Other authors applied mixed methods through time series and panel data [9]. In this work, we will analyze agrifood trade with a focus on exports and imports in three large sectors of agrifood.

### 2.1. Methods Issues

The question of the stationarity of the series and its effect on the analysis of structural breaks is a controversial and complex issue when analyzing structural breaks in time series. The problem is that most macroeconomic series contain a unit root, and, due to this fact, it is important to check. Syed and Zwick [25] highlight the stationarity of the series. However, these authors [25] point out that their analysis refers to structural breaks, and it would be more appropriate to study the unit root breakpoint and adopt the procedures of Glynn et al. [26]. The authors [13] conducted an extensive literature review on unit root tests that consider potential framework changes and applied the same methodology. Baldwin [27], in his work analyzing the price volatility in the U.S. milk sector, conducted a literature review of the different models applied and analyzed the issue of stationarity through the augmented Dickey–Fuller (ADF) test. In his work, there was strong evidence that the series is stationarity; an F-test was applied to test for structural breaks. Stationarity in the generation of data in series is an important condition for the properties of the produced estimators. Otherwise, we can produce spurious regressions that make no economic sense [28] (p. 216). Stationarity is a property that ensures that the means and variances are constant and independent of time, and the value of the covariance depends only on the distance between two periods and not on the current time in which it is calculated [29]. Economic series often show a trend or a pronounced seasonality component and temporal variations in variances [30]. Sometimes, this effect can be eliminated through transformations or the application of filters that allow working with seasonally stabilized series [31]. However, often, despite the transformation, the source of non-stationarity remains. Instabilities in the series can be caused by sources other than those mentioned above. These sources of instability are the events that can cause a structural break. It is called a structural break when a time series changes abruptly at one point in time. This change could involve a change in the mean or a change in the other parameters of the process that produce the series. Time series data can often contain a structural break due

to changes in policy or an exogenous economic shock. [32]. Works about unit roots in the presence of a structural break were developed by [33] and later treated by Perron [34].

Glynn et al. [26] carried out a bibliographic analysis considering the controversy of the stationarity of the series and the different models developed over time to test the existence of unit roots in the presence of structural breaks. Nevertheless, he said that even testing structural breaks when the series are non-stationary could affect the analysis of structural breaks [34]. The presence of breaks can bias the stationarity analysis and vice versa. The bias of test results in the series stationarity analysis is confirmed by Enders [28], highlighting the tests developed by [34] to test unit roots in the presence of structural changes. Hansen [35] carried out an analysis of the meaning of the structural break, identifying the break that occurs if some of the parameters of the series change at a given date as a structural break. It also refers to the fact that a break in the trend produces correlation properties in series like non-stationary series. The issue of structural breaks and stationarity is not simple and becomes more complex when referring to multiple structural breaks. [26] emphasizes this complexity and that considering the presence of a single break leads to a loss of information. Lumsdaine and Papell [36] developed the methodology of testing the unit root, an endogenous structural break for the alternative of two structural breaks, and it has been followed by other authors [37,38].

The presence of non-stationarity and the presence of structural breaks are factors that influence each other simultaneously [39]. As the controversy between stationarity and structural break in time series is an unresolved issue, we will perform the unit root contrast and evaluate the behavior of the series.

As the objective of this work is to analyze the structural breaks despite the stationarity analysis, it was decided to apply it to different models and tests. We apply it to the xtbreak model, to the stability analysis over time (CUSUM and OLS-CUSUM), and to the statistical tests for known and unknown structural breaks in beings at a level without differentiation regardless of their stationarity. The xtbreak package implements various tests for structural breaks in time series and panel data models [9] and can handle models of "pure" or "partial" structural change. Ditzen et al. [9] (p. 3) show the discussion of the model. The comparison between the xtbreak model and the statistics presented by the STATA 16 software is known as the sbknown stat, which performs a Wald test on time series by Know break data and the sbsingle stat for an unknown break [40]. All the tests are performed with STATA 16.1 software. To test the unit root, we employ the augmented Dickey–Fuller (ADF), the DF-GLS, and the Phillips–Perron unit-root checks. The lags elected have a main role in the test performance [41]. To choose the correct number of lags, we use the lags that minimize the Akaike Information Criterion (AIC), the Schwartz Bayesian information criterion (SBIC), and the lags that optimize Ng–Perron sequential test.

### 2.2. Structural Methods

Structural break test analysis can be performed in two different ways: testing a structural break at a known point or testing for breaks without knowing the breakpoint. The Chow test [42] is the best-known test for analyzing serial structure change and was introduced to analyze the existence of a structural break at a given point. To perform a Chow test, the sample must be separated into two periods, the breakpoint being the point of separation. After this division, the sample parameters are estimated, and the equality of the coefficients between the two samples is tested using the F statistic [30,41]. According to [43], this test has a weak capacity when the structure point is not known, and it is not robust when heteroscedasticity and autocorrelation are not guaranteed, existing according to the author's probability of rejecting the null hypothesis of stability due to autocorrelation issues in the errors or distinct variances. When you have time series, it is interesting to make a contrast between the null hypotheses of temporal homogeneity of the model versus the hypothesis of having produced a change. Therefore, tests called "recursive tests", also known as fluctuation tests, were developed. The two most representative tests are the CUSUM-type tests (Cumulative Sums of Standardized Residuals) and the OLS-CUSUM,

also known as the CUSUMSQ test [44,45]. In the STATA software, these tests are called CUSUM of recursive residuals and CUSUM of OLS residuals, respectively.

The CUSUM test examines the cumulative sum of recursive residuals, and under the null hypothesis, the cumulative sum of residuals will mean zero. The CUSUMSQ type looks at the cumulative sum of squares of the recursive residuals. The two tests are similar, but Turner [46] presents a study on the differences between the two tests. Kramer et al. [47] discuss limits for both tests, and some authors conclude that the version with OLS residues performs better in detecting breaks near the end of the series. Other authors have developed a literature review on these tests, their applications, relationships, and consistency [39,48,49].

When performing a time series regression, we assume that the coefficients are stable over time. These tests test this assumption and base their results on the case where the time series abruptly changes its behavior in ways not predicted by your model; that is, they test for structural breaks in the residuals. Vasco mentions that the problem with these tests is that they do not "detect breaks when the regressors are stationary and of zero means, because they are always orthogonal to any break" [50] (p. 20). Kramer et al. [47] raised important questions about the power of these tests and developed tests for dynamic systems. Several tests were developed to analyze stability and structural breaks. Some authors have conducted a review of the literature on the issue of unit roots and structural breaks [39,49,51,52]. For these points, it is worth mentioning the works of [53,54]. Banerjee et al. [54] pointed out that the endogenous structural break test is a sequential test that uses the complete sample and a different dummy variable for each possible break date. The model to be applied will influence the results, and the cutoff points depend on the model and its specifications. Since there is no consensus model, the empirical results must be read with care, and the interaction between the various tests must be considered.

Ditzen et al. [8,9] introduced a set of tests for the analysis of multiple breaks in time series and for models with panel data based on the work of Bai and Perron [55]. This procedure, called xtbreak, is integrated into the STATA software, and to perform it, it is not necessary to know the breakpoints. The main idea is that "if the model with the true break dates given a number of breaks has a smaller sum of squared residuals (SSR) than a model with incorrect break dates" (https://janditzen.github.io/xtbreak/#2-description, accessed on 12 December 2022).

As the controversy between stationarity and structural break in time series is an unresolved issue, we will perform the unit root contrast and evaluate the behavior of the series. However, as mentioned by Shikida et al. [39] and as previously mentioned, the presence of non-stationarity and the presence of structural breaks are factors that influence each other simultaneously. As the objective of this work is to analyze the breaks despite the stationarity analysis, it was decided to apply the xtbreak model [9], the stability analysis over time, and the statistical tests for known and unknown structural breaks in beings at a level without differentiation regardless of their stationarity. The comparison between the xtbreak model and the statistics presented by the STATA software as Wald tests for known and unknown break dates [40].

*2.3. Applied Data*

In this paper, we will analyze agrifood trade focusing on exports and imports in three major agrifood sectors: live animals and animal products (Animal); plant products (Vegetable) and prepared foods; beverages, alcoholic li/quids, and vinegar; tobacco and its manufactured substitutes; and other products (FB&T). These trade series are in accordance with the combined nomenclature—CN8 of Eurostat. To identify structural breaks in the long international trade series, the annual and the monthly series of the above-mentioned sectors were used since 2000 (Database: https://www.ine.pt/xportal/xmain?xpid=INE&xpgid=ine_bdc_tree&contexto=bd&selTab=tab2, accessed on 23 August 2023). The annual series covers the period from 2000 to 2021, and the monthly series analyzes the months from January 2000 to October 2022. The differences between the series at nominal and

real prices are more significant in 2022 when, according to the INE (2022), the Consumer Price Index (CPI) registered an average annual variation of 7.8%, significantly above the variation registered in 2021 as whole (1.3%) and in previous years. To deflate the annual series, we apply the Gross Domestic Product Deflator (base = 2015), and to deflate the monthly series, we apply the Harmonized Index of Consumer Prices (HICP, Base—2015). The need to use different deflators is related to the data provided by the National Statistics Institute of Portugal (INE-Instituto Nacional de Estatística).

## 3. Results

### 3.1. Stationarity Analysis of Trade Time Series

The previously mentioned unit root tests were carried out through the application of the STATA 16.1 Software in the annual and monthly series. The tests include the trend because the trend was a significant coefficient. According to the results, we cannot reject the fact that the annual and monthly trade series for agricultural products (Animal and Vegetable products) and the FB&T have a unit root, but the results are not consensual for all series and for all tests. In the monthly series, we cannot reject the stationarity for all tests and lags. In this work, we consider that the analysis of stationarity is important to better understand the results later, but the results were not presented because, as mentioned earlier, if there are breaks, these influence the results of the stationary. Applying the same methodology as [40], it was decided to refer to stationarity, but in the presentation of the results, the study on stationarity was omitted because it was decided to perform the tests on the level series; that is, it was not applied the differentiation in the series.

### 3.2. Analysis of Structural Breaks

In order to determine the regressions in which the stability tests will be applied, i.e., the CUSUM, OLS-CUSUM, and Wald and Supremum Wald tests (stat sbknown and sbsingle, respectively), it was necessary to determine the most appropriate number of lags to run the regressions. A pre-estimation analysis was performed on the multivariate time series models, estimating the Akaike Information Criterion (AIC), Hannan and Quinn Information Criterion (HQIC), and Schwarz Bayesian Information Criterion (SBIC) (Table 1). These results determined the lags of the regressions that were used in the subsequent stages of this study.

**Table 1.** Results of the VAR predestination tests for lag estimation.

| | Tests | AIC | HQIC | SBIC | AIC | HQIC | SBIC |
|---|---|---|---|---|---|---|---|
| | Type of series | | Annual | | | Monthly | |
| Exports | Animal | 2 | 2 | 2 | 4 | 4 | 4 |
| | Vegetables | 1 | 1 | 1 | 4 | 4 | 4 |
| | Food, Beverage, and Tobacco (FB&T) | 3 | 3 | 3 | 4 | 4 | 4 |
| Imports | Animal | 1 | 1 | 1 | 3 | 3 | 2 |
| | Vegetables | 3 | 3 | 3 | 3 | 3 | 3 |
| | Food, Beverage, and Tobacco (FB&T) | 2 | 2 | 2 | 3 | 3 | 2 |

After determining the number of lags indicated in the pre-estimation for each variable through the vector autoregressive (VAR) model, the result was confirmed by performing the regression for the maximum number of lags and carrying out a diagnosis of the regression by applying the "estat ic—Display information criteria" that calculates two information criteria used to compare models. In general, "smaller is better": given two models, the one with the smaller AIC fits the data better than the one with the larger AIC (STATA, https://www.stata.com/manuals16/restatic.pdf, accessed on 23 August 2023).

　　Once the lag number was confirmed, parameter stability tests were carried out, i.e., the Cumulative Sum test for parameter stability (Recursive and OLS-CUSUM). In Table 2, we can see the results for each series.

**Table 2.** Tests for parameter stability for annual and monthly series.

|  | Tests | Recursive CUSUM | OIS CUSUM | Recursive CUSUM | OIS CUSUM |
|---|---|---|---|---|---|
|  | Type of series | Annual | | Monthly | |
| Exports | Animal | 1.9782 | 1.9925 | 6.5568 | 6.8659 |
|  | Vegetables | 1.8382 | 1.9031 | 5.9350 | 6.8224 |
|  | FB&T | 2.2393 | 1.8667 | 7.2923 | 6.6261 |
| Imports | Animal | 1.6640 | 1.7734 | 4.5841 | 5.6884 |
|  | Vegetables | 1.4983 | 1.7465 | 4.3526 | 5.3816 |
|  | FB&T | 2.5150 | 1.7872 | 4.3855 | 5.6469 |

Note: Recursive: 1% critical value (1.143); 5% critical value (0.948); 10% critical value (0.850). OLS: 1% critical value (1.628); 5% critical value (1.358); 10% critical value (1.224).

　　The results of the absolute value Recursive CUSUM tests with critical values allow us to reject the null hypothesis of a constant mean at the critical 1% level because all values exceed the critical values of 1.1430 for all series: annual and monthly. We also tested the null hypotheses of parameter stability by applying the CUSUM of the OLS residuals, and the results in Table 2 show that we can reject the null hypothesis of parameter stability for all critical levels. The results provide information on the instability of regression models.

　　After this analysis, and according to the methodology presented, it was necessary to test the presence or absence of breakpoints and, in case they exist, the dates of the structural breakpoints. To analyze the structural breaks, the sequential test for multiple breaks at unknown breakpoints was applied by applying the "xtbreak" model, which provides a complete toolbox to analyze multiple structural breaks in a time series. It allows for determining the maximum number of sequential breaks, indicating the largest number of breaks for which the null hypothesis is rejected [9]. The breakpoints obtained by the sequential test were confirmed by the test of multiple breaks at known break dates proposed by the "xtbreak" model, which presents critical values of 1%, 5%, and 10% [55].

　　To complement the study, a structural break analysis was performed with an unknown break date. Wald's test was also performed to test whether the coefficients in the time regression vary over known break dates. The difference between testing a known breakpoint and an unknown breakpoint has to do with the critical values. In the case of testing unknown points, the critical values result from a non-normalized distribution. According to [9], these critical values are more realistic than when applying Chow's test. The test was also performed for each breakpoint separately. These breakpoints were then confirmed using the Supremum Wald test for a Known structural break date. The results were not always consistent across the different tests.

　　In Appendix A, we can see the results of the tests applying the xtbreak model and the Wald and Supermun Wald statistics for the annual international trade series by sector analyzed. In this table, the xtbreak command was applied to find out the number, structural breaks, dates, and confidence intervals. The xtbreak model sequentially determines the number of breaks [9]. The command for performing the sequential test for multiple breaks at unknown breakpoints in the STATA (16.1) program was xtbreak variable L. Additional tests were performed, testing the first assumption of the xtbreak command called Hypothesis 1 or Hypothesis A in which the hypothesis "Ho: no breaks versus H1; where the number of breaks under H1(s) where is specified by the researcher" [9]. The command for performing Hypothesis A was xtbreak test variables L.time, hypothesis(1) breaks(s), where "s" are the number of breaks. To test for multiple breaks at known break dates that are performing the regular Chow F-test, the command is "xtbreak test variable L.time, hypothesis (1) breakpoints (date list)" [8,9].

After analyzing the results of Hypothesis (1), Hypothesis (2) and Hypothesis (3) were tested. Hypothesis (2) or (B) tests the "H0: no breaks versus H1: $1 \leq s \leq$ smax breaks, where the maximum number of breaks in H1, smax, is specified by the researcher". Hypothesis (3) or (C) tests "H0: s breaks versus H1: s+1, where s is specified by the researcher". The sequential test continues until the null hypothesis is not rejected. The same methodology was applied to the annual and monthly time series.

### 3.2.1. Results for Annual Time-Series Trade

The results of the structural analysis on the annual export and import series and the Wald and Supermum Wald tests are presented in Appendix A. The Wald tests confirm the breaking points identified from the xtbreak model. There was also some instability (breaking points) in 2003–2004, 2009, and 2014–2017.

The results of the breakpoints analysis and the scatter plot of the annual time series (Figure 1) show a greater variation in imports than in exports. In the case of annual imports, the behavior is different for vegetable products, where changes in the behavior of the series are not verified using the xtbreak model but are verified for the Supermum Wald test.

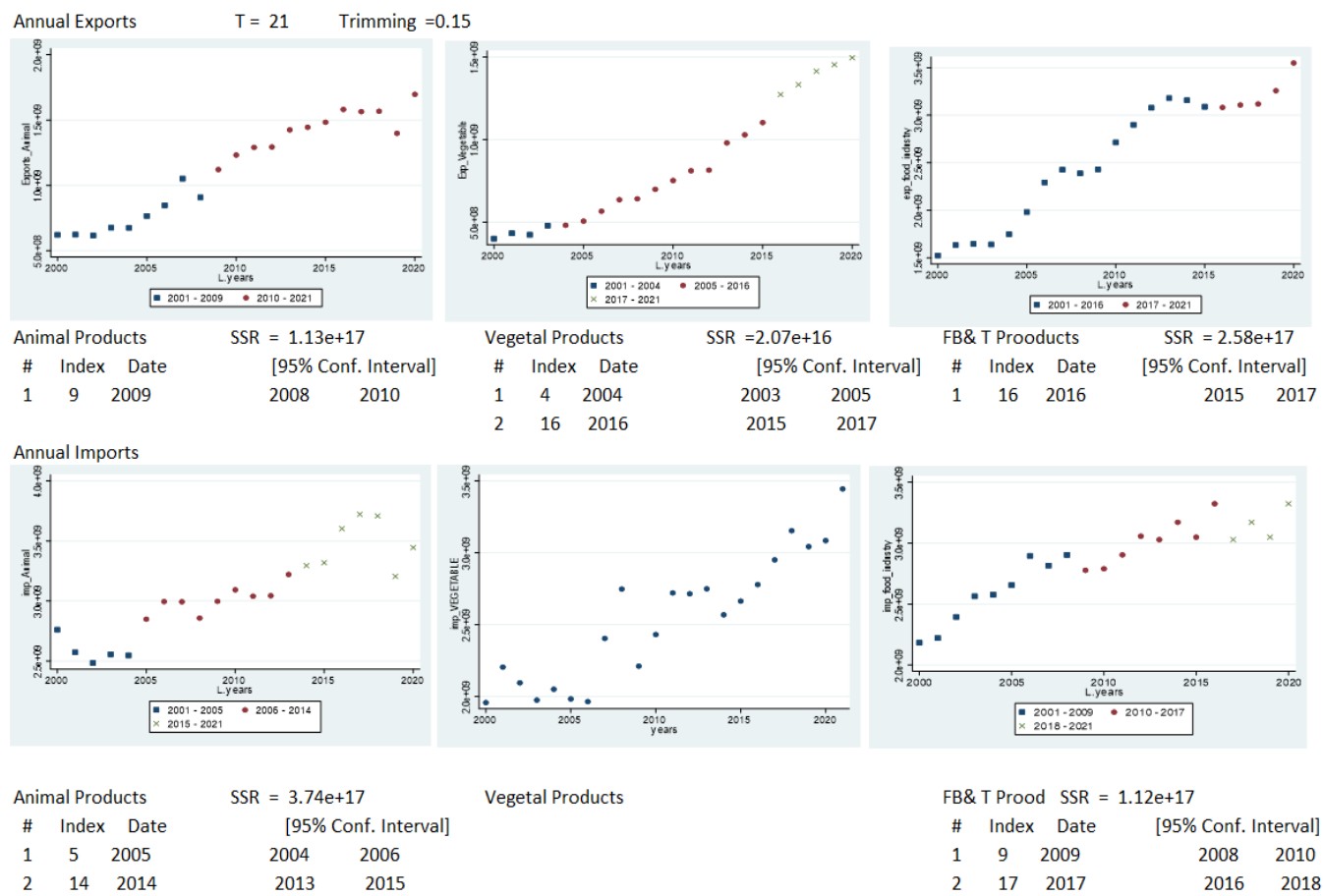

**Figure 1.** Scatter plot of annual time series and breakpoints with the xtbreak command.

In Appendix B, we present the additional tests, namely the hypotheses referred to by [13] as Hypotheses B and C and presented in the xtbreak as Hypotheses (2) and (3). The tests confirm the results obtained and are presented in Appendix A. In the imports of analog product animals, we cannot reject the null hypothesis of no breaks in relation to one, two, or three breaks.

Looking at exports, there are different behaviors between the three sectors: two breaks in plant products, coinciding with the disconnection of some CAP aid and the period of recovery of household disposable income; a break in the export of animal products, when

the global subprime crisis occurred; and a breaking point in processed foods coinciding with the recovery of household income.

In the case of imports, we have different dynamics across sectors, with no breaks in the behavior of imports of plant products but with two breaking points in animal products and processed products. These breaks in exports occur between different dates.

When comparing imports with exports, we find that there are also no coincidences in the breaking points within each sector of activity. A change in the behavior of imports does not change the behavior of exports, and vice versa. There are inter-annual variabilities, especially in the case of imports of plant products, but there are no coincidences in trend breaks in the behavior of the external market for agrifood products.

### 3.2.2. Results for Monthly Time-Series Trade

The monthly series, due to a large number of data points, allows better visualization of the series and the behavior of exports and imports of the sectors analyzed in the period between January 2000 and October 2022. In Appendix C, we can observe the results for breaks and breakpoints for the monthly time series of international trade according to the proposed methodology. The results are different for the exports and imports series. Figure 2 displays the breakpoints for the monthly data series.

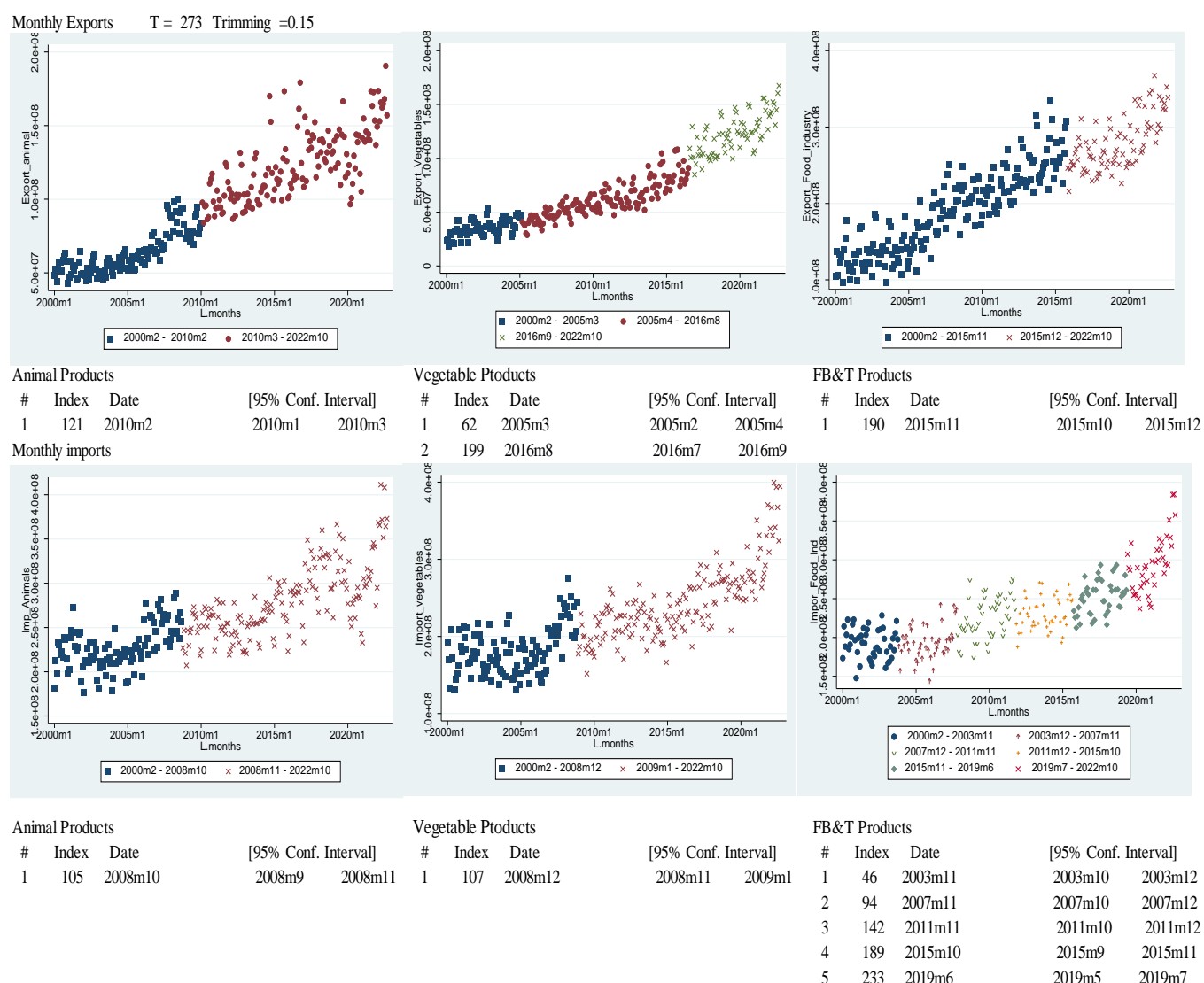

**Figure 2.** Scatter plots of monthly time series and breakpoints with the xtbreak command.

In the export series, the breakpoints fall in the confidence intervals of the annual analysis. There is a consistency between the monthly and annual results, with more clarity and greater temporal acuity in the results of the monthly analysis. For imports, we observe significant differences between the annual and monthly time-series breaking points.

Although the years 2015 and 2016 are breaking points for the vegetable sector and for the food, beverage, and tobacco industries (FB&T) exportations. The beginning of 2005 is another breaking point for vegetable production (the year 2004 in annual time-series analysis). For exports of animal products, the breaking point is in the second month of 2010 (in the year 2009 in the annual time-series analysis).

The evolution of agricultural exports is closely related to agricultural production, so we analyze the evolution of these two indicators. Also, taking into account the inter-annual variability of production, we will consider breaking points biannually. Thus, we consider the average values for the biennium's 2000–2001 (beginning of the series), 2004–2005, 2009–2010, 2015–2016 (the breaking points), and 2019–2020 (end of the series). The first breaking point coincides with the decoupling of direct aid from the CAP, having affected mainly crop production. The average annual growth rate (AAGR) of agricultural production was −0.4% between 2000 and 2001 and 2004 and 2005, and 1.2% between 2004 and 2005 and 2015 and 2016 (second breaking point) and slightly higher in the following period (1.4%) [19]. Agricultural exports have always had a high growth between the two periods considered, with growth rates of 8.1%, 9.9%, and 7.3%, respectively. The decoupling of aid can contribute to the increase in exports [56], and there was an acceleration in the period following the 2003 CAP reform. From 2015 to 2016, there was a slowdown in the growth rate, which may be associated with having reached an already very high level of specialization of agricultural production, with an effect on most exporting sectors, and is coincident with the recovery of the household income.

The behavior of imports of products of vegetable origin is similar to animal products, with a breaking point at the end of 2008 that corresponds to the global subprime financial crisis (the beginning of the sovereign debt, economic, and political crises). The FB&T series is much more unstable than the other two series, with five breaking points, a pace of every four years, and inexplicable economic or political explanations due to relevant factors such as the economic crisis, the COVID-19 pandemic, or changes in CAP.

Appendix D presents additional results that confirm the results previously obtained, with the exception of the series of imports of animal products where the test indicates only a break due to the breakdown of the regression constant.

## 4. Conclusions

Some volatility is observed in the international trade series, namely in the case of imports. In the case of exports, and if we analyze the monthly series, we verify the presence of between one and two structural breaks in the different sectors. The behavior of agricultural products (vegetable and animal products) is more coherent with the possible factors that may affect agricultural production and the consumption of agricultural goods, such as the withdrawal of some CAP aid (in 2004) and the crisis caused by the sovereign debt (global crisis at the end of 2008 and financial assistance between 2011 and 2014) which had a major impact on household consumption and, above all, on investment in general. The behavior of the food, beverage, and tobacco industries appears to be more closely associated with variations in income and economic cycles. It should be noted that the results of the analysis of the monthly series show greater consistency in the comparison between sectors and with the factors that may have more impact on the agrifood sector.

The most relevant period for the behavior of the series seems to be the period after the sovereign debt crisis that occurred in the Eurozone in 2008, and that had consequences in the following years. Portugal was subject to the process imposed by the International Monetary Fund (IMF) during the period 2011–2014. This period seems to have had some impact on international trade according to the breaking point in animal and vegetable imports. Imports of food products, beverages, and tobacco present five breakpoints, but in

some of the series, most cases, these refer to the October and November periods and may be related to other factors, namely the seasonality of the production, and not related to the sovereign debt crisis, COVID-19 pandemic, or CAP reform.

The COVID-19 pandemic caused the closure of sectors and borders and the reduction of trade worldwide, considering the economy as a whole, but such behavior seems to have had no significant impact on the international trade of the food products analyzed.

Agricultural production dynamics depend on several variables, namely global shocks such as war, climate change, macroeconomic policies, and pandemic disasters. In the case of the CAP effect, policies change slowly over time, considering that each change in CAP reform has a review process every seven years and there are transition periods. These effects will subsequently involve external trade, which will reflect the joint effect of these changes. Martinho [57] carried out a study about CAP effects on structural breaks in agricultural accounts in European countries. The results show that structural changes are due, in many cases, to changes in statistical information rather than to CAP reforms. The work [57] also reveals that the effects of CAP policies on European farms seem to take a few years. The effects of CAP alterations in the export and import series are difficult to define and detect, being, as mentioned, a hypothesis to be considered in the detected breaks.

Agriculture has a great dynamic effect on exportations during the two decades of the analysis, with a slight increase in the evolution of vegetable products after the decoupling of some CAP subsidies. The economic–financial crises did not affect the exportation behavior of vegetables and processed products. The animal products exportation had a breaking point at the end of 2010 but maintained a relevant trade dynamic, with an average annual rate of change of 8.0% between 2010 and 2020 [15]. The importations of animal and vegetable products revealed a change in their dynamics (at the end of 2008) but without a great change in the average annual rate of change and below the rate of growth of exportations (less than a half). The behavior of the Portuguese international agrifood trade markets reveals a resilience sector to the great impact of financial global crises and good adaptation to the decoupling of some CAP measures.

**Author Contributions:** Conceptualization, M.d.F.O.; methodology, M.d.F.O.; software, M.d.F.O.; validation, P.R.; formal analysis, M.d.F.O.; investigation, M.d.F.O. and P.R.; resources, M.d.F.O.; data curation, M.d.F.O.; writing—original draft preparation, M.d.F.O. and P.R.; writing—review and editing, P.R. All authors have read and agreed to the published version of the manuscript.

**Funding:** Research Centre for Natural Resources Environment and Society (CERNAS), Escola Superior Agrária de Coimbra.

**Institutional Review Board Statement:** Not applicable for studies not involving humans or animals.

**Data Availability Statement:** Not applicable.

**Conflicts of Interest:** The authors declare no conflict of interest.

## Appendix A

**Table A1.** Xtbreak results (Hypothesis A) and Wal and Supermum Wald tests (annual series).

| Annual Exports | Tests | Variable | Statistics | *p*-Value | Estimation of Breakpoints | | Bai and Perron Critical Values | | | (95% Conf. Interval) | |
|---|---|---|---|---|---|---|---|---|---|---|---|
| xtbreak | **Animal** hypotheses A | Sequential test for multiple breaks at unknown breakpoints Detected number of breaks and dates: | | | 1 break | 2 breaks | 1% - | 5% 1 | 10% 1 | | |
| | supW(tau) | H0: no break(s) vs. H1: 1 break(s) | 10.04 | | 2009 | | 12.29 | 8.58 | 7.04 | 2008–2010 | |
| | W(tau) | 1 break (2009) | 10.04 | 42.89 | | | | | | | |
| | | 1 break (2012) | 0.28 | 0.60 | | | | | | | |
| estat sbsingle | swald | **Animal (lag. 2)** | 91.62 | 0.00 | 2012 | | | | | | |
| estat sbknown | Wald test chi2(2) | 1 break (2009) | 42.89 | 0.00 | | | | | | | |
| xtbreak | **Vegetable** hypotheses A | Sequential test for multiple breaks at unknown breakpoints Detected number of breaks and dates: | | | 1 break | 2 breaks | 1% 1 | 5% 2 | 10% 2 | | |
| | supW(tau) | H0: no break(s) vs. H1: 1 break(s) | 42.69 | | 2016 | | 12.29 | 8.58 | 7.04 | 2003–2005 | |
| | supW(tau) | H0: no break(s) vs. H1: 2 break(s) | 46.61 | | | 2004; 2016 | 9.36 | 7.22 | 6.28 | 2015–2017 | |
| | W(tau) | 1 break (2016) | 42.69 | 0.00 | | | | | | | |
| | W(tau) | 1 break (2004) | 20.16 | 0.00 | | | | | | | |
| | W(tau) | 1 break (2016; 2004) | 46.61 | 0.00 | | | | | | | |
| | W(tau) | 1 break (2015) | 29.09 | 0.00 | | | | | | | |
| estat sbsingle | swald | **Vegetable (1)** | 71.40 | 0.00 | 2015 | | | | | | |
| estat sbknown | Wald test chi2(2) | 1 break (2016) | 58.93 | 0.00 | | | | | | | |
| | | 1 break (2004) | 5.28 | 0.02 | | | | | | | |
| | | 2 break(s) (2004 2016) | 81.40 | 0.00 | | | | | | | |
| xtbreak | **FB&T** hypotheses A | Sequential test for multiple breaks at unknown breakpoints Detected number of breaks and dates: | | | 1 break | 2 breaks | 1% 1 | 5% 1 | 10% 1 | | |
| | supW(tau) | H0: no break(s) vs. H1: 1 break(s) | 24.58 | | 2016 | | 12.29 | 8.58 | 7.04 | 2015–2017 | |
| | W(tau) | 1 break (2016) | 24.58 | 0.00 | | | | | | | |
| | W(tau) | 1 break (2010) | 4.53 | 0.05 | | | | | | | |
| estat sbsingle | swald | **FB&T (lag 3)** | 67.53 | 0.00 | 2010 | | | | | | |
| estat sbknown | Wald test chi2(2) | Break date (2016) | 27.64 | 0.00 | | | | | | | |

**Table A1.** *Cont.*

| Annual Imports | Tests | Variable | Statistics | *p*-value | Estimation of breakpoints | | Bai and Perron Critical Values | | | (95% Conf. Interval) | |
|---|---|---|---|---|---|---|---|---|---|---|---|
| xtbreak | **Animal** | Sequential test for multiple breaks at unknown breakpoints | | | 1 break | 2 breaks | 1% | 5% | 10% | | |
| | hypotheses A | Detected number of breaks and dates: | | | | | - | 2 | 2 | | |
| | supW(tau) | H0: no break(s) vs. H1: 2 break(s) | 1.24 | | 2013 | | 12.29 | 8.58 | 7.04 | 2004–2006 | |
| | | H0: no break(s) vs. H1: 2 break(s) | 3.14 | | | 2005–2014 | 9.36 | 7.22 | 6.28 | 2013–2015 | |
| | W(tau) | 1 break (2005) | 1.22 | 0.280 | | | | | | | |
| | W(tau) | 1 break (2014) | 1.18 | 0.290 | | | | | | | |
| | W(tau) | 2 break(s) (2005 2014) | 3.14 | 0.060 | | | | | | | |
| | W(tau) | 1 break (2015) | 0.93 | 0.350 | | | | | | | |
| estat sbsingle | swald | **Animal (lag. 1)** | 36.06 | 0.000 | 2015 | | | | | | |
| estat sbknown | Wald test chi2(2 | 1 break (2005) | 8.81 | 0.003 | | | | | | | |
| | | 1 break (2014) | 31.96 | 0.000 | | | | | | | |
| | | 2 break(s) (2005–2014) | 43.45 | 0.000 | | | | | | | |
| xtbreak | **Vegetable** | Sequential test for multiple breaks at unknown breakpoints | | | 1 break | 2 breaks | 1% | 5% | 10% | | |
| | hypotheses A | Detected number of breaks and dates: | | | | | - | - | - | | |
| | W(tau) | 1 break (2010) | 0.21 | 0.650 | | | | | | | |
| | swald | **Vegetable (lag 3)** | 41.86 | 0.000 | 2010 | | | | | | |
| | **FB&T** | Sequential test for multiple breaks at unknown breakpoints | | | 1 break | 2 breaks | 1% | 5% | 10% | | |
| | hypotheses A | Detected number of breaks and dates: | | | | | - | 2 | 2 | | |
| | supW(tau) | H0: no break(s) vs. H1: 2 break(s) | 16.36 | 0 | | 2009–2017 | 9.36 | 7.22 | 6.28 | 2008–2010 | |
| | W(tau) | 1 break (2009) | 0.80 | 0.38 | | | | | | 2016–2018 | |
| | W(tau) | 1 break (2017) | 11.02 | 0.00 | | | | | | | |
| | W(tau) | 2 break(s) (2009 2017) | 16.36 | 0.00 | | | | | | | |
| estat sbsingle | swald | **FB&T (lag 2)** | 50.95 | 0.00 | 2009 | | | | | | |
| estat sbknown | Wald test chi2(2) | Break date (2017) | 12.80 | 0.00 | | | | | | | |

## Appendix B

**Table A2.** Additional results for Hypothesis B (2) and Hypothesis C (3) (annual series).

| Annual Exports | | Hypotheses | | Test | Statitic | 1% | 5% | 10% | Analysis |
|---|---|---|---|---|---|---|---|---|---|
| Animal | B | H0: no break(s) vs. H1: 1 ≤ s ≤ 1 break(s) | max = 1 | UDmax(tau) | 10.04 | 12.37 | 8.88 | 7.46 | Null hypotheses of no breaks against the alternative of up to 1 break. The null hypothesis is rejected at the 5% level. |
| | C | H0: 0 vs. H1: 1 break(s) | s = 0 | F(s+1|s) | 10.04 | 12.29 | 8.58 | 7.04 | Null hypotheses of no breaks against 1 break. We can reject the null hypothesis at the 5% level and accept one break at the 5% level. |
| | C | H0: 1 vs. H1: 2 break(s) | s = 1 | F(s+1|s) | 6.16 | 13.89 | 10.13 | 8.51 | Null hypotheses of 1 break against 2 breaks. We cannot reject the null hypothesis. |
| Vegetable | B | H0: no break(s) vs. H1: 1 ≤ s ≤ 2 break(s) | s max = 2 | UDmax(tau) | 46.61 | 12.37 | 8.88 | 7.46 | Null hypotheses of no breaks against the alternative of up to 2 breaks. The null hypothesis is rejected at the 1% level. |
| | C | H0: 0 vs. H1: 1 break(s) | s = 0 | F(s+1|s) | 42.69 | 12.29 | 8.58 | 7.04 | Null hypotheses of 0 breaks against 1 break. We can reject the null hypothesis at the 1% level and accept one break at the 1% level. |
| | C | H0: 1 vs. H1: 2 break(s) | s = 1 | F(s+1|s) | 11.54 | 13.89 | 10.13 | 8.51 | Null hypotheses of 0 breaks against 2 breaks. We can reject the null hypothesis at the 5% level and accept two breaks at the 5% level. |
| | C | H0: 2 vs. H1: 3 break(s) | s = 2 | F(s+1|s) | 2.75 | 14.8 | 11.14 | 9.41 | Null hypotheses of 2 breaks against 3 breaks. We cannot reject the null hypothesis. |
| FB&T | B | H0: no break(s) vs. H1: 1 ≤ s ≤ 2 break(s) | s max = 2 | UDmax(tau) | 24.58 | 12.37 | 8.88 | 7.46 | Null hypotheses of no breaks against the alternative of up to 2 breaks. The null hypothesis at the 1% level is rejected. |
| | C | H0: 0 vs. H1: 1 break(s) | s = 0 | F(s+1|s) | 24.58 | 12.2 | 8.58 | 7.04 | Null hypotheses of 0 breaks against 1 break. We can reject the null hypothesis at the 1% level and accept one break at the 1% level. |
| | C | H0: 1 vs. H1: 2 break(s) | s = 1 | F(s+1|s) | 5.28 | 13.89 | 10.13 | 8.51 | Null hypotheses of 1 break against 2 breaks. We cannot reject the null hypothesis. |
| Annual Imports | | | | | | | | | |
| Animal | B | H0: no break(s) vs. H1: 1 ≤ s ≤ 2 break(s) | s max = 2 | UDmax(tau) | 3.14 | 12.37 | 8.88 | 7.46 | Null hypothesis of no breaks against the alternative of up to 1 break. We cannot reject the null hypothesis. |
| | C | H0: 0 vs. H1: 1 break(s) | s = 0 | F(s+1|s) | 1.24 | 12.29 | 8.58 | 7.04 | Null hypothesis of 0 breaks against 1 break. We cannot reject the null hypothesis. |
| Vegetable | B | H0: no break(s) vs. H1: 1 ≤ s ≤ 1 break(s) | s max = 1 | UDmax(tau) | 3.18 | 12.37 | 8.88 | 7.46 | Null hypotheses of no breaks against the alternative of up to 1 break. We cannot reject the null hypothesis. |
| | C | H0: 0 vs. H1: 1 break(s) | s = 0 | F(s+1|s) | 1.27 | 12.29 | 8.58 | 7.04 | Null hypotheses of 0 breaks against 1 break. We cannot reject the null hypothesis. |

**Table A2.** *Cont.*

| Annual Exports | | Hypotheses | | Test | Statitic | 1% | 5% | 10% | Analysis |
|---|---|---|---|---|---|---|---|---|---|
| FB&T | B | H0: no break(s) vs. H1: 1 ≤ s ≤ 2 break(s) | s max = 2 | UDmax(tau) | **16.36** | 12.37 | 8.88 | 7.46 | Null hypothesis of no breaks against the alternative of up to 2 breaks. The null hypothesis at the 1% level is rejected. |
| | C | H0: 0 vs. H1: 1 break(s) | s = 0 | F(s+1\|s) | **11.02** | 12.29 | 8.58 | 7.04 | Null hypotheses of 0 breaks against 1 break. We can reject the null hypothesis at the 5% level and accept one break at the 5% level. |
| | C | H0: 1 vs. H1: 2 break(s) | s = 1 | F(s+1\|s) | **13.84** | 13.89 | 10.13 | 8.51 | Null hypothesis of 1 break against 2 breaks. We can reject the null hypothesis at the 5% level and accept 2 breaks at the 5% level. |
| | C | H0: 2 vs. H1: 3 break(s) | s = 2 | F(s+1\|s) | **4.39** | 14.80 | 11.14 | 9.41 | Null hypothesis of 2 breaks against 3 breaks. We cannot reject the null hypothesis. |

## Appendix C

**Table A3.** Xtbreak results (Hypothesis A) and Wal and Supermum Wald tests (monthly series).

| Monthly Exports | Tests | Variable | Statistics | *p*-Value | Estimation of Breakpoints | | | Bai and Perron Critical Values | | | (95% Conf. Interval) |
|---|---|---|---|---|---|---|---|---|---|---|---|
| xtbreak | **Animal** | Sequential test for multiple breaks at unknown breakpoints | | | 1 break | 2 breaks | | 1% | 5% | 10% | |
| | Hypothesis A | Detected number of breaks and dates: | | | | | | | 1 | 1 | |
| | supW(tau) | H0: no break(s) vs. H1: 1 break(s) | 10.87 | | 2010m2 | | | 12.29 | 8.58 | 7.04 | 2010m1 2010m3 |
| | W(tau) | 1 break (2010m2) | 10.87 | 0.00 | | | | | | | |
| | | 1 break (2011m1) | 4.68 | 0.03 | | | | | | | |
| estat sbsingle | Swald | **Animal (lag. 4)** | 633.94 | 0.00 | 2011m1 | | | | | | |
| estat sbknown | Wald test chi2 | 1break (2010m2) | 586.19 | 0.00 | | | | | | | |
| xtbreak | **Vegetable** | Sequential test for multiple breaks at unknown breakpoints | | | 1 break | 2 breaks | | 1% | 5% | 10% | |
| | Hypothesis A | Detected number of breaks and dates: | | | | | | 2 | 2 | 2 | |
| | supW(tau) | H0: no break(s) vs. H1: 1 break(s) | 77.63 | | 2004m6 | | | 12.29 | 8.58 | 7.04 | 2005m2 2005m4 |
| | supW(tau) | H0: no break(s) vs. H1: 2 break(s) | 89.29 | | | 2005m3; 2016m8 | | 9.36 | 7.22 | 6.28 | 2016m7 2016m9 |
| | W(tau) | 1 break (2004m6) | 77.63 | 0.00 | | | | | | | |
| | W(tau) | 1 break (2005m3) | 83.35 | 0.00 | | | | | | | |
| | W(tau) | 1 break (2016m8) | 149.93 | 0.00 | | | | | | | |
| | W(tau) | 2 break(s) (2005m3; 2016m8) | 89.29 | 0.00 | | | | | | | |



**Table A3.** *Cont.*

| Monthly Exports | Tests | Variable | Statistics | *p*-Value | Estimation of Breakpoints | | | | Bai and Perron Critical Values | | | (95% Conf. Interval) |
|---|---|---|---|---|---|---|---|---|---|---|---|---|
| | W(tau) | 1 break (2015m1) | 76.68 | 0.00 | | | | | | | | |
| estat sbsingle | Swald | **Vegetable (4)** | 777.04 | 0.00 | 2015m1 | | | | | | | |
| estat sbknown | Wald test chi2 | 1 break (2004m6) | 89.90 | 0.00 | | | | | | | | |
| | | 1 break (2005m3) | 113.96 | 0.00 | | | | | | | | |
| | | 1 break (2016m8) | 677.68 | 0.00 | | | | | | | | |
| | | 2 break(s) (2005m3; 2016m8) | 1061.75 | 0.00 | | | | | | | | |
| xtbreak | **FB&T** | Sequential test for multiple breaks at unknown breakpoints | | | 1 break | 2 breaks | | | 1% | 5% | 10% | |
| | Hypothesis A | Detected number of breaks and dates | | | | | | | 1 | 1 | 1 | |
| | supW(tau) | H0: no break(s) vs. H1: 1 break(s) | 25.58 | | 2015m11 | | | | 12.29 | 8.58 | 7.04 | 2015m10; 2015m12 |
| | W(tau) | 1 break (2015m11) | 25.58 | 0.00 | | | | | | | | |
| | W(tau) | 1 break (2011m9) | 0.07 | 0.79 | | | | | | | | |
| estat sbsingle | Swald | **FB&T (4)** | 504.11 | 0.00 | 2011m9 | | | | | | | |
| estat sbknown | Wald test chi2 | Break date (2015m11) | 215.21 | 0.00 | | | | | | | | |

| Monthly Imports | Tests | Variable | Statistics | *p*-value | Estimation of breakpoints | | | | Bai and Perron Critical Values | | | (95% Conf. Interval) |
|---|---|---|---|---|---|---|---|---|---|---|---|---|
| xtbreak | **Animal** | Sequential test for multiple breaks at unknown breakpoints | | | 1 break | 2 breaks | 3 breaks | 4 breaks | 1% | 5% | 10% | |
| | Hypothesis A | Detected number of breaks and dates: | | | | | | | 4 | 1 | 1 | |
| | supW(tau) | H0: no break(s) vs. H1: 1 break(s) | 12.09 | | 2008m10 | | | | 12.29 | 8.58 | 7.04 | 2008m9 2008m11 |
| | | H0: no break(s) vs. H1: 2 break(s) | 12.88 | | | 2003m5; 2011m8 | | | 9.36 | 7.22 | 6.28 | |
| | | H0: no break(s) vs. H1: 3 break(s) | 11.85 | | | | 2003m5; 2008m5; 2011m9 | | 7.6 | 5.96 | 5.21 | |
| | | H0: no break(s) vs. H1: 4 break(s) | 10.06 | | | | | 2003m5; 2006m1; 2011m8; 2017m2 | 6.19 | 4.99 | 4.41 | |
| | W(tau) | 1 break (2008m10) | 12.09 | 0.00 | | | | | | | | |
| | | 1 break (2003m5) | 7.73 | 0.01 | | | | | | | | |
| | | 1 break (2011m8) | 7.21 | 0.01 | | | | | | | | |
| | | 1 break (2008m5) | 9.73 | 0.00 | | | | | | | | |
| | | 1 break (2006m1) | 0.02 | 0.88 | | | | | | | | |
| | | 1 break (2017m2) | 27.57 | 0.00 | | | | | | | | |
| | | 2 break(s) (2003m5; 2011m8) | 12.88 | 0.00 | | | | | | | | |

**Table A3.** *Cont.*

| Monthly Imports | Tests | Variable | Statistics | *p*-value | Estimation of breakpoints | | | | | Bai and Perron Critical Values | | | (95% Conf. Interval) |
|---|---|---|---|---|---|---|---|---|---|---|---|---|---|
| estat sbsingle estat sbknown | swald Wald test chi2 | 3 break(s) (2003m5; 2008m5; 2011m9) | 11.85 | 0.00 | | | | | | | | | |
| | | 4 break(s) (2003m5; 2006m1; 2011m8; 2017m2) | 11.73 | 0.00 | | | | | | | | | |
| | | 1 break (2015m6) | 13.66 | 0.00 | | | | | | | | | |
| | | **Animal (lag. 3)** | 293.88 | 0.00 | 2015m6 | | | | | | | | |
| | | 1 break (2008m10) | 148.78 | | | | | | | | | | |
| | | 2 break(s) (2003m5; 2011m8) | 184.14 | 0.00 | | | | | | | | | |
| | | 3 break(s) (2003m5; 2008m5; 2011m9) | 206.40 | 0.00 | | | | | | | | | |
| | | 4 break(s) (2003m5; 2006m1; 2011m8; 2017m2) | 462.27 | 0.00 | | | | | | | | | |
| | **Vegetable** | Sequential test for multiple breaks at unknown breakpoints | | | 1 break | 2 breaks | | | | 1% | 5% | 10% | |
| | Hypothesis A | Detected number of breaks and dates | | | | | | | | 1 | 1 | 1 | |
| | supW(tau) | H0: no break(s) vs. H1: 1 break(s) | 17.47 | | 2008m12 | | | | | 12.29 | 8.58 | 7.04 | 2008m11 2009m1 |
| | W(tau) | 1 break (2008m12) | 17.47 | | 0 | | | | | | | | |
| | W(tau) | 1 break (2017m3) | 26.63 | | 0 | | | | | | | | |
| estat sbsingle estat sbknown | swald | **Vegetable (3)** | 271.70 | 0.00 | 2017m3 | | | | | | | | |
| | Wald test chi2 | 1 break (2008m12) | 155.52 | 0.00 | | | | | | | | | |
| | **FB&T** | Sequential test for multiple breaks at unknown breakpoints | | | 1 break | 2 breaks | 3 breaks | 4 breaks | 5 breaks | 1% | 5% | 10% | |
| | Hypothesis A | Detected number of breaks and dates: | | | | | | | | 5 | 5 | 5 | |
| | supW(tau) | H0: no break(s) vs. H1: 1 break(s) | 18.69 | | 2004m12 | | | | | 12.29 | 8.58 | 7.04 | 2003m10 2003m12 |
| | | H0: no break(s) vs. H1: 2 break(s) | 22.71 | | | | 2003m11; 2011m11 | | | 9.36 | 7.22 | 6.28 | 2007m10 2007m12 |
| | | H0: no break(s) vs. H1: 3 break(s) | 15.15 | | | | | 2003m11; 2009m11; 2013m10 | | 7.6 | 5.96 | 5.21 | 2011m10 2011m12 |
| | | H0: no break(s) vs. H1: 4 break(s) | 13.48 | | | | | | 2003m11; 2009m11; 2013m10; 2019m6 | 6.19 | 4.99 | 4.41 | 2015m9 2015m11 |
| | | H0: no break(s) vs. H1: 5 break(s) | 11.09 | | | | | | 2003m11; 2007m11; 2011m11; 2015m10; 2019m6 | 4.91 | 3.91 | 3.47 | 2019m5 2019m7 |

**Table A3.** *Cont.*

| Monthly Imports | Tests | Variable | Statistics | *p*-value | Estimation of breakpoints | Bai and Perron Critical Values | (95% Conf. Interval) |
|---|---|---|---|---|---|---|---|
| | W(tau) | 1 break (2004m12) | 18.69 | 0.00 | | | |
| | | 1 break (2003m11) | 19.26 | 30.43 | | | |
| | | 1 break (2009m11) | 1.84 | 0.18 | | | |
| | | 1 break (2013m10) | 1.10 | 0.29 | | | |
| | | 1 break (2019m6) | 30.43 | 30.43 | | | |
| | | 1 break (2015m10) | 5.41 | 0.02 | | | |
| | | 1 break (2007m11) | 51.91 | 0.00 | | | |
| | | 2 break(s) (2003m11; 2011m11) | 22.71 | 0.00 | | | |
| | | 3 break(s) (2003m11; 2009m11; 2013m10) | 15.15 | 0.00 | | | |
| | | 4 break(s) (2003m11; 2009m11; 2013m10; 2019m6) | 13.48 | 0.00 | | | |
| | | 5 break(s) (2003m11; 2007m11; 2011m11; 2015m10; 2019m6) | 11.09 | 0.00 | | | |
| | | 1 break (2016m8) | 12.37 | 0.00 | | | |
| estat sbsingle | Swald | **FB&T (3)** | 250.00 | 0.00 | 2016m8 | | |
| estat sbknown | Wald test chi2 | 1 break (2004m12) | 76.35 | 0.00 | | | |
| | | 2 break(s) (2003m11; 2011m11) | 227.89 | 0.00 | | | |
| | | 3 break(s) (2003m11; 2009m11; 2013m10) | 314.82 | 0.00 | | | |
| | | 4 break(s) (2003m11; 2009m11; 2013m10; 2019m6) | 479.89 | 0.00 | | | |
| | | 5 break(s) (2003m11; 2007m11; 2011m11; 2015m10; 2019m6) | 556.79 | 0.00 | | | |

## Appendix D

**Table A4.** Additional results of Hypothesis B (2) and Hypothesis C (3) (monthly series).

| Monthly Exports | | Hypotheses | | Test Statistics | | 1% | 5% | 10% | Analysis |
|---|---|---|---|---|---|---|---|---|---|
| Animal | B | H0: no break(s) vs. H1: $1 \leq s \leq 1$ break(s) | s max = 1 | UDmax(tau) | **10.87** | 12.37 | 8.88 | 7.46 | Null hypothesis of no breaks against the alternative of up to 1 break. The null hypothesis is rejected at the 5% level. |
| | C | H0: 0 vs. H1: 1 break(s) | s = 0 | F(s+1\|s) | **10.87** | 12.29 | 8.58 | 7.04 | Null hypothesis of no breaks against 1 break. We can reject the null hypothesis at the 5% level and accept one break at the 5% level. |
| | C | H0: 1 vs. H1: 2 break(s) | s = 1 | F(s+1\|s) | **6.19** | 13.89 | 10.13 | 8.51 | Null hypothesis of 1 break against 2 breaks. We cannot reject the null hypothesis. |
| Vegetable | B | H0: no break(s) vs. H1: $1 \leq s \leq 2$ break(s) | s max = 2 | UDmax(tau) | **89.29** | 12.37 | 8.88 | 7.46 | Null hypotheses of no breaks against the alternative of up to 2 breaks. The null hypothesis is rejected at the 1% level. |
| | C | H0: 0 vs. H1: 1 break(s) | s = 0 | F(s+1\|s) | **77.63** | 12.29 | 8.58 | 7.04 | Null hypotheses of 0 breaks against 1 break. We can reject the null hypothesis at the 1% level and accept one break at the 1% level. |
| | C | H0: 1 vs. H1: 2 break(s) | s = 1 | F(s+1\|s) | **57.65** | 13.89 | 10.13 | 8.51 | Null hypothesis of 0 breaks against 2 breaks. We can reject the null hypothesis at the 5% level and accept two breaks at the 1% level. |
| | C | H0: 2 vs. H1: 3 break(s) | s = 2 | F(s+1\|s) | **7.14** | 14.8 | 11.14 | 9.41 | Null hypothesis of 2 breaks against 3 breaks. We cannot reject the null hypothesis. |
| FB&T | B | H0: no break(s) vs. H1: $1 \leq s \leq 2$ break(s) | s max = 2 | UDmax(tau) | **25.58** | 12.37 | 8.88 | 7.46 | Null hypothesis of no breaks against the alternative of up to 2 breaks. The null hypothesis at the 1% level is rejected. |
| | C | H0: 0 vs. H1: 1 break(s) | s = 0 | F(s+1\|s) | **25.58** | 12.29 | 8.58 | 7.04 | Null hypothesis of 0 breaks against 1 break. We can reject the null hypothesis at the 1% level and accept one break at the 1% level. |
| | C | H0: 1 vs. H1: 2 break(s) | s = 1 | F(s+1\|s) | **6.58** | 13.89 | 10.13 | 8.51 | Null hypothesis of 1 break against 2 breaks. We cannot reject the null hypothesis. |
| Monthly Imports | | | | | | | | | |
| Animal | B | H0: no break(s) vs. H1: $1 \leq s \leq 4$ break(s) | s max =4 | UDmax(tau) | **12.88** | 12.37 | 8.88 | 7.46 | Null hypothesis of no breaks against the alternative of up to 4 breaks. The null hypothesis at the 1% level is rejected. |
| | C | H0: 0 vs. H1: 1 break(s) | s = 0 | F(s+1\|s) | **12.09** | 12.29 | 8.58 | 7.04 | Null hypotheses of 0 breaks against 1 break. We can reject the null hypothesis at the 5% level and accept one break at the 5% level. |
| | C | H0: 1 vs. H1: 2 break(s) | s = 1 | F(s+1\|s) | **2.19** | 13.89 | 10.13 | 8.51 | Null hypothesis of 1 break against 2 breaks. We cannot reject the null hypothesis. |

**Table A4.** *Cont.*

| Monthly Exports | | Hypotheses | | Test Statistics | | 1% | 5% | 10% | Analysis |
|---|---|---|---|---|---|---|---|---|---|
| Vegetable | B | H0: no break(s) vs. H1: $1 \leq s \leq 2$ break(s) | s max = 2 | UDmax(tau) | **23.63** | 12.37 | 8.88 | 7.46 | Null hypothesis of no breaks against the alternative of up to 2 breaks. The null hypothesis at the 1% level is rejected. |
| | C | H0: 0 vs. H1: 1 break(s) | s = 0 | F(s+1 \| s) | **17.47** | 12.29 | 8.58 | 7.04 | Null hypothesis of 0 breaks against 1 break. We can reject the null hypothesis at the 1% level and accept one break at the 1% level. |
| | C | H0: 1 vs. H1: 2 break(s) | s = 1 | F(s+1 \| s) | **4.88** | 13.89 | 10.13 | 8.51 | Null hypothesis of 1 break against 2 breaks. We cannot reject the null hypothesis. |
| FB&T | B | H0: no break(s) vs. H1: $1 \leq s \leq 5$ break(s) | s max =5 | UDmax(tau) | **22.71** | 12.29 | 8.58 | 7.04 | Null hypotheses of no breaks against the alternative of up to 5 breaks. The null hypothesis at the 1% level is rejected. |
| | C | H0: 0 vs. H1: 1 break(s) | s = 0 | F(s+1 \| s) | **18.69** | 12.29 | 8.58 | 7.04 | Null hypothesis of 0 breaks against breaks. We can reject the null hypothesis at the 1% level and accept one break at the 1% level. |
| | C | H0: 1 vs. H1: 2 break(s) | s = 1 | F(s+1 \| s) | **22.9** | 13.89 | 10.13 | 8.51 | Null hypothesis of 1 break against 2 breaks. We can reject the null hypothesis at the 1% level and accept 2 breaks at the 1% level. |
| | C | H0: 2 vs. H1: 3 break(s) | s = 2 | F(s+1 \| s) | **16.27** | 14.80 | 11.14 | 9.41 | Null hypothesis of 2 breaks against 3 breaks. We can reject the null hypothesis at the 1% level and accept 3 breaks at the 1% level. |
| | C | H0: 3 vs. H1: 4 break(s) | s = 3 | F(s+1 \| s) | **20.13** | 15.28 | 11.83 | 10.04 | Null hypothesis of 3 breaks against 4 breaks. We can reject the null hypothesis at the 1% level and accept 4 breaks at the 1% level. |
| | C | H0: 4 vs. H1: 5 break(s) | s = 4 | F(s+1 \| s) | **35.78** | 15.76 | 12.25 | 10.58 | Null hypothesis of 4 breaks against 5 breaks. We can reject the null hypothesis at the 1% level and accept 5 breaks at the 1% level. |
| | C | H0: 5 vs. H1: 6 break(s) | s = 5 | F(s+1 \| s) | **38.18** | 16.27 | 12.66 | 11.03 | Null hypothesis of 5 breaks against 6 breaks. We can reject the null hypothesis at the 1% level and accept 6 breaks at the 1% level. |

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
