# Peer review of "Portuguese Agrifood Sector Resilience: An Analysis Using Structural Breaks Applied to International Trade"

_agriculture, doi:10.3390/agriculture13091699_

Round 1
Reviewer 1 Report
1. The drafting of the paper is very complex and plenty of repetitions. The wording must be simplified.
2. I would suggest a more simple presentation, stressing on the objectives, previous literature, main hypothesis, justification of the chosen method, data and findings and discussion.
3. The article does not help to understand breaks. It only detects them. Previous formulation of hypothesis and the discussion would help to understand structural breaks.
4. The presentation of objectives in the introduction is not very clear. Objectives must be explicitly stated.
5. Citations to previous literature on food trade in Portugal are not many.
6. What is the paper’s added value to existing literature? Is it the issue of stationarity? If this is the case, it should be clearly stated at the introduction.
7. In different parts of the paper, authors argue in favour of xtbreak commmand, referring it to as a toolbox to analyse structural breaks. Avoid repetitions and instead explain which specific commands were used and why.
9. Annexes must be shortened and simplified. It would be advisable to have a summary table of the main tests or simplify the presentation.
10. Is the analysis applied to only 6 time series? 3 for exports and 3 for imports? What is the statistical analysis adding to a simple observation of the time series or introduction of dummy variables responding to external shocks?
11. In the Discussion, it is interesting that authors reflect on the sources of the shocks implying structural changes. What I don’t see is the underlying theory behind those changes. Did the authors considered a structural regression model, with changing parameters? Perhaps some of the changes are not so structural and rather respond to a change in policy variables (e.g. CAP payments).
12. Which are the policy implications of the analysis?
The quality of the English language is very poor. I strongly recommend the article to be checked by a native speaker.
Author Response
suggest a more simple presentation, stressing on the objectives, previous literature, main hypothesis, justification of the chosen method, data and findings and discussion.
R: OK, review seeking to address the suggestion.
- The article does not help to understand breaks. It only detects them. Previous formulation of hypothesis and the discussion would help to understand structural breaks.
R: OK, we believe that the revised manuscript responds to the relevant commentary.
- The presentation of objectives in the introduction is not very clear. Objectives must be explicitly stated.
R: OK, revision made to explicitly the objectives of the study.
- Citations to previous literature on food trade in Portugal are not many.
R: The references on the international trade of the agri-food sector are the data of the National Institute of Statistics worked by the Ministry of Agriculture.
- What is the paper’s added value to existing literature? Is it the issue of stationarity? If this is the case, it should be clearly stated at the introduction.
R: The question of stationarity is a complementary analysis to the identification of outbreaks and the Wald and Supermun Wald tests. This complementarity makes the analysis of the behaviour of time series (imports and exports) more robust. The authors apply the bibliography to support the analysis of the importance of stationarity and the methodology followed in this work.
The main contribution to the existing literature is to demonstrate the resilience of the Portuguese agri-food sector to global shocks, and to demonstrate the applicability of a robust methodology for analysing the behaviour of time series. The authors consider that this work contributed by applying a recent method in the analysis of time series and its application to analyze events such as the economic crisis and the covid in international trade series in a sector considered a priority in the pandemic crisis.
- In different parts of the paper, authors argue in favour of xtbreak commmand, referring it to as a toolbox to analyse structural breaks. Avoid repetitions and instead explain which specific commands were used and why.
R: The commands used and why are indicated in the reviewed manuscript.
- Annexes must be shortened and simplified. It would be advisable to have a summary table of the main tests or simplify the presentation.
R: The Figures 1 and 2 summarise the results presented in the annexes. We consider important to keep all the results in the annexes because they demonstrate the robustness of the results obtained.
- Is the analysis applied to only 6 time series? 3 for exports and 3 for imports? What is the statistical analysis adding to a simple observation of the time series or introduction of dummy variables responding to external shocks?
R: Yes, there are only six time series, three for exports and three for imports.
Statistical analysis complements simple observations. According Hansen (2001, p. 118) (https://pubs.aeaweb.org/doi/pdf/10.1257/jep.15.4.117m). “The econometrics of structural change looks for systematic methods to identify structural breaks. In the past 15 years, the most important contributions to this literature include the following three innovations: 1) Tests for a structural break of unknown timing; 2) Estimation of the timing of a structural break; and 3) Tests to distinguish between a random walk and broken time trends. These three innovations have dramatically altered the face of applied time series econometrics. We discuss these three topics in turn and use U.S. labor productivity data to illustrate their applicability”. The goal is to identify the timing of a structural break with innovative xtbreak model and compared with classic Wald test.
- In the Discussion, it is interesting that authors reflect on the sources of the shocks implying structural changes. What I don’t see is the underlying theory behind those changes. Did the authors considered a structural regression model, with changing parameters? Perhaps some of the changes are not so structural and rather respond to a change in policy variables (e.g. CAP payments).
R: What we want to demonstrate is the ability to respond to large shocks by maintaining the dynamics of export growth.
- Which are the policy implications of the analysis?
R: The results show the capacity of the agri-food sector to respond to external shocks, which highlights the importance of maintaining policies to support the agri-food sector, especially in terms of investment and innovation, because agents have the capacity to respond to adjustment needs.
Comments on the Quality of English Language
The quality of the English language is very poor. I strongly recommend the article to be checked by a native speaker.
- The authors improved the English language.

Reviewer 2 Report
Upon reviewing this manuscript, I find it quite intriguing; however, there are several aspects that require improvement. The specific points that need attention are outlined below:
· The abstract should be rewritten to ensure clarity. It is important to provide information regarding the source of data, the methodology employed, the obtained results, the contribution to the field, the limitations, and the significance of this research.
· In the keywords section, it is unclear what "xtbreak" represents. Prior to discussing any topic, please explain the meaning of "xtbreak."
· The introduction lack’s structure and could benefit from the inclusion of background information.
· The absence of a literature review is noticeable, as the manuscript immediately delves into the methodology. Restructuring is necessary to rectify this omission.
· The title mentions a data series; however, no analysis of the series is presented. It is advisable to incorporate an analysis of the extensive dataset.
· All abbreviations used should be properly defined for clarity and ease of understanding.
· The entire manuscript requires restructuring, and missing sections such as limitations, contributions, and a literature review should be incorporated.
In order to enhance the manuscript's quality and coherence, addressing these points will be crucial.
Author Response
regarding the source of data, the methodology employed, the obtained results, the contribution to the field, the limitations, and the significance of this research.
R: OK, revised the summary according to the suggestions.
- In the keywords section, it is unclear what "xtbreak" represents. Prior to discussing any topic, please explain the meaning of "xtbreak."~
R: OK. “xtbreak” is a Stata module for estimating and testing structural breaks in time series and panel data. That is why Stata was added to the keyword “xtbreak” (the statistic software·
The introduction lack’s structure and could benefit from the inclusion of background information.
R: OK, revised the introduction to respond to the comment. We consider that the basic information is related to agri-food external trade and is already included in the introduction.
- The absence of a literature review is noticeable, as the manuscript immediately delves into the methodology. Restructuring is necessary to rectify this omission.
- Ok, revised the manuscript seeking to respond to the comment. Included new references. We believe it is important to keep the deeper review on the methodology because it is the crucial part of our analysis. There is very little literature on the effects of the 2008 global shocks and the pandemic on the Portuguese agri-food sector.
- The title mentions a data series; however, no analysis of the series is presented. It is advisable to incorporate an analysis of the extensive dataset.
R: Ok. Revised title. Extensive data analysis would get confused with the analysis of results so we think the best option is the one followed by the authors.
- All abbreviations used should be properly defined for clarity and ease of understanding.
R: OK, we think that with the revision of the manuscript this issue is resolved.

Round 2
Reviewer 1 Report
The work has improved significantly and addressed most of my previous comments.
The work has improved significantly and addressed most of my previous comments.